# Peer review of "VEGF/Neuropilin Signaling in Cancer Stem Cells"

_ijms, 2019, doi:10.3390/ijms20030490_

Round 1
Reviewer 1 Report
I read the review with great interest and I have not suggestions for the author.
I found the article much well designed and wrote.
The only concern regards the Figure 1 that effectively shows highly generic.
Author Response
We have added more detail to Figure 1.
Reviewer 2 Report
This is a comprehensive review summarized functional roles of VEGF/Neuropilins signaling in tumorigenesis including cancer stem cells self-renewal and survival, extends beyond angiogenesis and vascular permeability. In addition to VEGF receptor tyrosine kinases, the neuropilins mediate the effects of VEGF on CSCs via growth factor receptors and integrins. Therefore, active VEGF/NRP signaling axis in cancer stem cells acts as a novel therapeutic target that confer resistance to standard chemotherapy. This review is a great contribution to the cancer research field and will bring great interest for the readers.
Epithelial-to-endothelial transition, a phenotypic switch of cancer stem cells, plays critical roles in cancer progression and metastasis (two recent publications below). Does VEGF/Neuropilins signaling involve in EET? It will be helpful to add some discussion in this review.
1. Oncotarget, 2017, Vol. 8, (No. 18), pp: 30502-30510. Epithelial-to-endothelial transition and cancer stem cells: two cornerstones of vasculogenic mimicry in malignant tumors.
2. NATURE COMMUNICATIONS | (2018) 9:4315. Targeting SPINK1 in the damaged tumour microenvironment alleviates therapeutic resistance.
Author Response
We appreciate the mention of EMT and have included a paragraph to address this issue. However, the EET, which the reviewer mentions, is not appropriate for the scope of this review. The papers mentioned, although interesting, are also not relevant to this review.
Reviewer 3 Report
There are a few typing errors, such as
lane 181, VGFR2 (VEGFR2);
lane 232, spice (splice).
I have read Dr. Mercurio's previous papers cited in the references. This review is constructed by using them as major scaffold.
I think NRP2 is more reasonable as the cancer stem cell (CSC) target because its expression was much higher in CSC than cancer cell (CC).
NRP1 is expressed higher in normal epithelial than transformed cells.
Figure 1 is a very expressive model. The author may describe it more deeply, if possible.
Author Response
We have corrected the typos mentioned. We respectfully disagree with the comments on NRP expression in normal epithelia and tumors based on the literature that we cite.
We have described Figure 1 in more depth.